# The Dipterose of Black Soldier Fly (*Hermetia illucens*) Induces Innate Immune Response through Toll-Like Receptor Pathway in Mouse Macrophage RAW264.7 Cells

**DOI:** 10.3390/biom9110677

**Published:** 2019-10-31

**Authors:** Muhammad Fariz Zahir Ali, Takashi Ohta, Atsushi Ido, Chiemi Miura, Takeshi Miura

**Affiliations:** 1Graduate School of Agriculture, Ehime University, 3-5-7, Tarumi, Matsuyama, Ehime 790-8566, Japan; fariz290392@gmail.com (M.F.Z.A.); ido@agr.ehime-u.ac.jp (A.I.); c.miura.6u@it-hiroshima.ac.jp (C.M.); 2South Ehime Fisheries Research Center, Ehime University, 1289-1, Funakoshi, Ainan, Ehime 798-4292, Japan; ot1059@gmail.com; 3Department of Global Environment Studies, Faculty of Environmental Studies, Hiroshima Institute of Technology, 2-1-1 Miyake, Saeki-ku, Hiroshima 731-5193, Japan

**Keywords:** polysaccharide, insect, innate immunity, macrophage, Toll-like receptor

## Abstract

In our study, a novel bioactive polysaccharide was identified in the larvae of the black soldier fly (BSF) (*Hermetia illucens*) as a molecule that activates the mammalian innate immune response. We attempted to isolate this molecule, which was named dipterose-BSF, by gel-filtration and anion-exchange chromatography, followed by nitric oxide (NO) production in mouse RAW264.7 macrophage cells as a marker of immunomodulatory activity. Dipterose-BSF had an average molecular weight of 1.47 × 10^5^ and consisted of ten monosaccharides. Furthermore, in vitro assays demonstrated that dipterose-BSF enhanced the expression of proinflammatory cytokines and interferon β (IFNβ) in RAW264.7 cells. The inhibition of Toll-like receptor 2 (TLR2) and 4 (TLR4) significantly attenuated NO production by dipterose-BSF, indicating that dipterose-BSF stimulates the induction of various cytokines in macrophages via the TLR signaling pathway. This observation was analogous with the activation of nuclear factor kappa B in RAW264.7 cells after exposure to dipterose-BSF. Our results suggest that dipterose-BSF has immunomodulatory potential through activating the host innate immune system, which allows it to be a novel immunomodulator for implementation as a functional food supplement in poultry, livestock, and farmed fish.

## 1. Introduction

The emergence of polysaccharides as therapeutic agents has attracted worldwide attention in recent decades [1]. Distinct pharmacological properties, such as anti-tumor, immunomodulatory, anti-diabetes, anticoagulant, antiviral, antioxidant, and other activities were found to be associated with bioactive polysaccharides [2,3]. Moreover, polysaccharides are relatively safe, non-toxic, and biodegradable, which make them suitable for application in agriculture and aquaculture fields [4]. Thus, utilization of bioactive polysaccharides might serve as a promising alternative to conventional therapeutic methods, such as antibiotic treatment and vaccination.

Functional polysaccharides from insects have remained unnoticed compared to those from plants, fungi, and bacteria. Previously, three novel acidic polysaccharides were successfully identified from *Bactrocera cucurbitae* pupae [5], *Antheraea yamamai* pupae [6], and *Bombyx mori* pupae [7]. These bioactive polysaccharides induce the activation of innate immune responses in mouse macrophage RAW264.7 cells via Toll-like receptor (TLR) signaling pathway, leading to the secretion of various proinflammatory cytokines and prevention of infectious diseases. According to these findings, insects may have a remarkable capacity to produce bioactive polysaccharides with therapeutic properties.

The black soldier fly (BSF) (*Hermetia illucens*) is an insect commonly found in the tropics and warm temperate regions [8]. The larvae and pre-pupae are massively produced and used as protein sources of feed for poultry and fish culture [9]. The feeding activity of BSF is restricted to the larval stage; for this reason, they are not harmful to crops and human habitats [10]. Moreover, recent studies have revealed that the dietary intake of BSF enhances the immune activities in yellow catfish (*Pelteobagrus fulvidraco*) [11] and broiler chicks [12], suppresses the growth of infectious bacteria in weaned piglets [13], increases serum lysozyme activity in yellow catfish *(Pelteobagrus fulvidraco*) [11], and increases the expression of stress response-related genes in Atlantic salmon (*Salmo salar*) [14]. Dietary inclusion of BSF was reported to alter gut microbiota, including increased bacterial diversity to improve gut health in rainbow trout (*Oncorhynchus mykiss*) [15,16], increased levels of *Carnobacterium* genus, which is known to act as a probiotic for microbial communities in rainbow trout (*Oncorhynchus mykiss*) [17] and poultry [18]. Some studies concluded that BSF might possess bioactive compounds, such as chitin. Based on these facts, together with previous findings of insect-derived polysaccharides, we further investigated the possibility of a novel immune activator in BSF.

Here, a novel polysaccharide, named dipterose of BSF (dipterose-BSF), was isolated and purified from the BSF larvae using a series of purification methods. The molecular weight and monosaccharide composition of dipterose-BSF were characterized. The potency of dipterose-BSF as an immune activator was assessed by evaluating the nitric oxide (NO)-producing activity and expression of proinflammatory cytokines in mouse macrophage-derived RAW264.7 cell line. In addition, the possible signaling pathways involved in the activation of RAW264.7 cells were elucidated. Therefore, our study shows the potency of BSF as a source of bioactive polysaccharides that can induce immune responses, as well as explores its immunomodulatory mechanisms.

## 2. Materials and Methods

### 2.1. Preparation of BSF Larvae Extract

BSF (*Hermetia illucens*) were harvested and collected from the wild in Ainan, Ehime Prefecture, Japan. The method for collecting BSF eggs and rearing the larvae was adapted from Nakamura et al. [19]. The reared larvae were autoclaved for 20 min at 121 °C to inhibit endogenous enzymes and stored at –30 °C until further use. The frozen BSF larvae were homogenized using a grinder. The resulting BSF larvae meal was diluted with ten volumes of ultrapure water and mixed gently for 24 h at 4 °C. The supernatant was obtained by centrifugation at 20,000× *g* for 1 h and concentrated to about one-tenth of its volume by evaporation in a water bath at 50 °C.

### 2.2. Purification of a Bioactive Polysaccharide from BSF Larvae Extract

Bioactive polysaccharides from BSF were purified as described previously [5,6,7]. In brief, the BSF larvae extract was added to four volumes of 100% (*v/v*) ethanol and mixed gently overnight at 4 °C to precipitate the polysaccharides. The polysaccharides were precipitated by centrifugation, washed consecutively three times with 80% ethanol, and dried under decompression. The precipitate was dissolved in 20 mM Tris-HCl (pH 8.0), shaken overnight at 4 °C, and centrifuged at 20,000× *g* to separate the remaining low-molecular-weight compounds. The crude polysaccharides were then obtained after removing the precipitate.

The crude polysaccharides were injected into a HiPrep 26/60 Sephacryl S-500HR gel-filtration chromatography column (GE Healthcare, Chicago, IL, USA) pre-equilibrated with 20 mM Tris-HCl (pH 8.0) and eluted with the same solution at a flow rate of 1.3 mL/min. The eluates were collected automatically and monitored by the phenol-sulfuric acid method with glucose standard for total carbohydrate determination [20]. To evaluate its potent immunomodulatory activity, each eluted fraction was assayed for NO production in RAW264.7 cells. Fractions exhibiting NO-producing activity in RAW264.7 cells were then collected and precipitated by the addition of four volumes of 100% (*v/v*) ethanol overnight at 4 °C. The resulting precipitate was separated by centrifugation and diluted in 20 mM Tris-HCl (pH 8.0).

Anion-exchange chromatography on a HiPrep DEAE FF 16/10 column (GE Healthcare) was used for further purification. The resulting polysaccharides from gel-filtration chromatography were loaded to the above column pre-equilibrated with 20 mM Tris-HCl (pH 8.0) and eluted with the same solution. The fractions were produced with a linear gradient of NaCl from 0 to 1.0 M at a flow rate of 2.0 mL/min. The eluted fractions were collected automatically and assayed for NO production as mentioned above. The positive fractions in the NO production assay were pooled and precipitated by the addition of four volumes of 100% (*v/v*) ethanol overnight at 4 °C. The precipitate collected by centrifugation was lyophilized to obtain the bioactive polysaccharide.

### 2.3. Determination of Molecular Weight of the BSF Polysaccharide

The molecular weight of the BSF polysaccharide was measured by gel-filtration chromatography using high-performance liquid chromatography (Hitachi, Chiyoda, Japan) as described previously [5,6,7]. In brief, the purified polysaccharide (1 mg/mL) was dissolved in 0.2 M phosphate buffer (PB) and filtered through a 0.22 μm membrane. The sample solution was then applied to a Showdex SB-807 HQ size-exclusion chromatographic column (Showa Denko K.K., Minato, Japan), eluted with 0.2 M PB (pH 7.5) at a flow rate of 0.5 mL/min, and detected by a refractive index detector. The column temperature was kept at 35 °C. Pullulans of different molecular weights (P-5, P-10, P-20, P-50, P-100, P-200, P-00, P-800, and P-2500) were passed through the column, and a standard curve was prepared by plotting their retention time against the logarithms of their respective molecular weights. The molecular weight was calculated according to the calibration equation from the standard curve of pullulans.

### 2.4. Determination of the Monosaccharide Composition of the BSF Polysaccharide

Determination of the monosaccharide composition of the BSF polysaccharide was performed in accordance with our previous studies [5,6,7]. In brief, the purified polysaccharide (100 μg) was hydrolyzed by the addition of 2 M trifluoroacetic acid at 100 °C for 16 h. The resulting products were then evaporated using N_2_ stream and converted to alditol acetates by sequential NaBH_4_ reduction and acetylation with Ac_2_O-pyridine (1:1, *v/v*) following a published method by Sassaki et al. [21]. Gas chromatography-mass spectrometry (GC-MS) analyses were performed on an Agilent 7890A gas chromatography system (Agilent Technologies, Santa Clara, CA, USA) equipped with an HP5 capillary column (30 m × 0.35 mm × 0.25 μm) and connected to a JEOL MS-1050Q instrument (JEOL, Akishima, Japan). The column temperature was initially set at 100 °C, increased gradually to 180 °C at the rate of 10 °C, held for 5 min, and then elevated to 320 °C at 10 °C and finally kept for 5 min. Helium was used as the carrier gas and maintained at 1.0 mL/min. The molar ratio of monosaccharides was calculated by the area normalization method with myo-inositol used as the internal standard.

### 2.5. Cell Culture

The mouse macrophage-derived RAW264.7 cell line was obtained from a Cell Bank (RIKEN BioResource Center, Tsukuba, Japan). Cells were cultured in minimum essential medium (MEM) (Life Technologies, Grand Island, NE, USA) supplemented with 10% fetal bovine serum, 0.1 mM non-essential amino acids, 100 U/mL penicillin, and 100 μg/mL streptomycin. Cells were maintained at 37 °C under humidified air with 5% CO_2_.

### 2.6. Measurement of NO Production

The nitrite concentration in the supernatant of the culture medium of RAW264.7 cells was measured using the Griess reagent system kit (Promega, Madison, WI, USA) in accordance with the manufacturer’s instructions. Briefly, cells were seeded at 1 × 10^6^ cells/well in a 96-well plate, pre-incubated for 90 min, and stimulated with 100 ng/mL of lipopolysaccharide (LPS), varying concentrations of larvae extract or purified polysaccharide in the cultured medium for 24 h at 37 °C. The absorbance of culture medium supernatant was identified at optical density (OD) 540 nm in microplate reader. The nitrite levels in the culture medium were then measured based on the equation from a NaNO_2_ standard curve.

### 2.7. TLR2 and TLR4 Blocking Experiment

RAW264.7 cells were plated in a 96-well plate (1 × 10^6^ cells/well) and then pre-incubated for 1 h at 37 °C (5% CO2) with a neutralizing antibody against mouse TLR2 (InvivoGen, San Diego, CA, USA) or mouse anti-human TLR4 (10 μg/mL) (Invitrogen, Carlsbad, CA, USA) or with an isotype control antibody (InvivoGen). The purified polysaccharide from the BSF larvae (100 ng/mL) or LPS (100 ng/mL) was then applied to the wells and incubated for 20 h at 37 °C under a humidified atmosphere with 5% CO_2_. Nitrite concentrations in the supernatant of the culture medium were measured using the Griess reagent system kit as described above.

### 2.8. Isolation of Total RNA and Real-Time PCR

Total RNA was isolated from RAW264.7 cells using RNeasy Plus Universal Mini Kit (Qiagen, Hilden, Germany) following the manufacturer’s protocol. For complementary DNA synthesis, 500 ng total RNA was reverse-transcribed using the QuantiTect Reverse Transcription kit (Qiagen). Real-time PCR analysis was performed on a Bio-Rad CFX96 Real-Time PCR detection system (Bio-Rad Laboratories, Hercules, CA, USA) using SsoFast EvaGreen Supermix (Bio-Rad Laboratories). Primers were designed and synthesized by Eurofins based on the nucleotide sequence published by Xia et al. [22]. All experiments were run in triplicate, and the average of the four values was used for calculation. For normalization, glyceraldehyde-3-phosphate dehydrogenase was used as an endogenous reference. The relative expression of the target gene to the reference gene was calculated using the 2^–ΔΔCt^ method.

### 2.9. Immunoblot Analysis

To prepare cell lysates, cells (1 × 10^6^ cells/well) were washed three times with ice-cold phosphate buffered saline (PBS) and lysed by sonicating for 30 s with the addition of 200 μL of lysis buffer (137 mM NaCl, 10 mM phosphate, 2.7 mM KCl, pH 7.4, 1 mM EDTA, and 1% Triton X-100) supplemented with Halt protease inhibitor cocktail (Thermo Fisher Scientific, Rockford, IL, USA). The cell lysates were centrifuged at 20,400× *g* for 20 min at 4 °C to discard the insoluble material. For cytosolic and nuclear protein extraction, NE-PER nuclear and cytoplasmic extraction reagents (Thermo Fisher Scientific) were used in accordance with the manufacturer’s protocol. Aliquots of 10–20 μg of denatured protein were separated electrophoretically on 12.5% or 15% sodium dodecyl sulfate-polyacrylamide gels (Atto, Taito, Japan) and transferred to polyvinylidene difluoride membranes (Millipore Sigma, Burlington, VT, USA) by electroblotting. The membranes were blocked with tris buffered saline (TBS) (20 mM Tris-HCl, pH 7.5, 500 mM NaCl) containing 0.5% blocking reagent (Roche, Mannheim, Germany) for 1 h at 25 °C and washed three times with TBS containing 0.1% Tween-20 (TTBS, pH 7.5). The membranes were then incubated at 4 °C overnight with the following primary antibodies: rabbit anti-mouse inducible nitric oxide synthase (iNOS), rabbit anti-human nuclear factor-kappa B (NF-κB) p65 (Santa Cruz Biotechnology, Santa Cruz, CA, USA), or mouse anti-human NF-κB inhibitor alpha (IκBα) (Cell Signaling Technology, Danvers, MA, USA) pre-diluted (1:1000) in immunoreaction enhancer solution (Can Get Signal Solution 1; Toyobo, Osaka, Japan). Goat anti-human lamin A/C (Santa Cruz Biotechnology) and mouse anti-mouse α-tubulin (Sigma-Aldrich, St. Louis, MO, USA) were used as a loading control for nuclear protein and total protein, respectively. After washing with TTBS, membranes were incubated with anti-rabbit, anti-mouse, or anti-goat IgG alkaline phosphatase diluted (1:5000) in immunoreaction enhancer solution (Can Get Signal Solution 2; Toyobo) for 1 h at room temperature. Signal development was observed using CDP-star detection reagent (GE Healthcare) and visualized using the ImageQuant LAS 4000 (Fujifilm, Minato, Japan).

### 2.10. Statistical Analysis

All data are expressed as the mean ± standard error of mean (SEM). Statistical analysis was performed with one-way analysis of variance (ANOVA) followed by Tukey’s honestly significant difference (Tukey’s HSD) post hoc test using Minitab (State College, PA, USA) statistical software. Differences with *p* < 0.05 were considered statistically significant.

## 3. Results

### 3.1. Isolation of a Bioactive Polysaccharide from the BSF Larvae

In order to assess the potency of the BSF larvae for innate immune activation, we first tested the NO production levels in mouse macrophage-derived RAW264.7 cell line treated with various concentration of BSF larvae extract. As shown in Figure 1, the addition of BSF larvae extract induced NO production in RAW264.7 cells in the same manner as that of LPS, a widely known immunomodulator. Based on this result, we attempted to further identify the potency of BSF larvae by isolating its innate immune-activator compound.

After a series of processing steps, including water extraction, ethanol precipitation, and centrifugation, the crude polysaccharides were isolated from the BSF larvae. The purification of the immunomodulatory compound from BSF larvae extracts was conducted by using it as a marker of NO-producing activity in RAW264.7 cells. Positive fractions in the NO production assay were separated and collected using gel filtration and anion-exchange chromatography column on a fast protein liquid chromatography (FPLC) system. As shown in Figure 2A, several fractions containing sugars were found to stimulate NO production in RAW264.7 cells. In accordance with our previous findings of insect-derived polysaccharides [5], we further selected one NO-stimulating fraction that possessed the highest molecular weight, which was then further purified by anion-exchange chromatography on a HiPrep DAEA FF 16/10 column (GE Healthcare) eluted with a linear gradient of NaCl ranging from 0 to 1.0 M. In vitro analysis of the resulting fractions revealed that a single polysaccharide-comprising peak stimulated NO production in RAW264.7 cells (Figure 2B).

High-performance liquid chromatography on a Showdex SB-807 HQ column was performed to measure the molecular weight of the bioactive polysaccharide. According to the regression equation from the standard curve made by different Pullulan P-series standards, the average molecular weight of the bioactive polysaccharide was estimated to be 1.47 × 10^5^. We named this novel polysaccharide “dipterose of black soldier fly” (dipterose-BSF) after our previous finding of dipterose purified from the pupae of melon fly (*Bactrocera cucurbitae*).

### 3.2. Identification of the Monosaccharide Composition of Dipterose-BSF

The monosaccharide composition of dipterose-BSF was identified by GC-MS analysis with acid hydrolysis and acetylation techniques (Figure 3). According to the monosaccharide standards, dipterose-BSF was mainly composed of ten monosaccharides: l-rhamnose, l-fucose, l-arabinose, d-xylose, d-glucuronic acid, d-mannose, d-glucose, d-galactose, *N*-acetyl-d-glucosamine, and *N*-acetyl-d-galactosamine with a relative molar ratio of 8.1, 2.2, 3.1, 1.5, 5.6, 4.3, 17.8, 23.8, 21.7, and 11.9, respectively.

### 3.3. Effects of Dipterose-BSF on the Activation of Innate Immune System In Vitro

To investigate the immunomodulatory activity of dipterose-BSF, the levels of nitrite were measured in RAW264.7 cells treated with various concentrations of dipterose-BSF. As expected, the levels of nitrite were significantly increased in dipterose-BSF-treated RAW264.7 cells in a dose-dependent manner (Figure 4A). Surprisingly, the addition of 100 ng/mL dipterose-BSF induced NO production in RAW264.7 cells in a similar manner as that of LPS, a potent immunomodulator. In macrophages, including RAW264.7 cells, NO is synthesized by iNOS. To evaluate whether the expression of iNOS protein is affected by stimulation with dipterose-BSF, we monitored the expression level of iNOS protein in RAW264.7 cells by immunoblot analysis (Figure 4B). Treatment with indicated dose of dipterose-BSF resulted in a dose-dependent increase in the level of iNOS protein.

It is widely accepted that innate immune response is stimulated by the recognition of pattern recognition receptors (PRRs), including TLRs, to induce NO production and various proinflammatory cytokines via the activation of downstream signaling molecules. To investigate whether dipterose-BSF stimulates cytokine expression in macrophages, we analyzed the expression of the proinflammatory cytokines, interferon regulatory factor 3 (*IRF3*) and *IRF7*, in RAW264.7 cells treated with various concentrations of dipterose-BSF using real-time PCR analysis (Figure 5). The mRNA expression of several proinflammatory cytokines, such as tumor necrosis factor-alpha (*TNF-α*), interleukin (*IL*)-6, and *IL-1β* mRNA, which was mediated by a myeloid differentiation primary response protein 88 (MyD88)-dependent pathway, was significantly increased at 6 h after dipterose-BSF treatment in a dose-dependent manner. In addition, dipterose-BSF also significantly evoked the mRNA expression level of *IRF7* and interferon β (*IFNβ*), which was mediated by an MyD88-independent pathway at 6 h after the addition of dipterose-BSF in a dose-dependent manner. Unexpectedly, dipterose-BSF did not affect the expression of *IRF3* mRNA in RAW264.7 cells.

### 3.4. Effects of TLR2 and TLR4 Inhibition on the NO-Producing Activity of Dipterose-BSF In Vitro

Our findings revealed that dipterose-BSF induces proinflammatory cytokine and NO production via MyD88-dependent and -independent pathways through the recognition of TLRs in RAW264.7 cells. Therefore, to investigate whether dipterose-BSF induces NO production in RAW264.7 cells via TLRs, we applied neutralizing antibodies against TLR2 and TLR4 to inhibit their function, and thus determine their effect on NO production by RAW264.7 cells (Figure 6). Both antibodies significantly attenuated dipterose-BSF-induced NO production, indicating that dipterose-BSF stimulates the induction of the innate immune system via the TLR signaling pathway.

### 3.5. Effects of Dipterose-BSF on the TLR Signaling Pathway

We found that dipterose-BSF stimulates the innate immune system in RAW264.7 cells via the TLR signaling pathway. NF-κB is an important transcription factor functioning downstream of the TLR signal transduction pathway and is mainly stored in the cytoplasm in an inactive form by binding with IκBα, the inhibitor NF-κB. The degradation of IκBα leads to the nuclear translocation and activation of NF-κB. To investigate whether NF-κB regulates dipterose-BSF activities, we monitored the degradation of IκBα and the translocation of NF-κB to the nucleus after treatment with dipterose-BSF. The application of dipterose-BSF to RAW264.7 cells induced rapid degradation of IκBα in the same manner as that of LPS (Figure 7A). The degradation of IκBα occurred at 15 min post-treatment with dipterose-BSF, but recovered to basal levels after 60 min. On the other hand, the nuclear translocation of NF-κB was found to be increased at 15 min post-treatment with dipterose-BSF, suggesting the correlation with the result of IκBα degradation (Figure 7B).

## 4. Discussion

Polysaccharides, along with proteins, lipids, and polynucleotides, are fundamental biological macromolecules that play an important role in the growth and development of a living organism [1]. Natural polysaccharides from different sources, including plant, fungi, seaweed, and animal have been widely studied and reported to stimulate immune activities through direct or indirect interaction with immune system components [1,2,3,23]. Recently, insects are not only promising feedstuffs for livestock or cultured fish, but are known to exhibit functions for optimizing animal health [24]. Dietary inclusion of dried housefly (*Musca domestica*) pupa [25] and defatted yellow mealworm (*Tenebrio molitor*) [26] has been shown to increase disease resistance against pathogenic *Edwardsiella tarda* in red seabream (*Pagrus major*). Yellow mealworm is also known to increase the enzyme activities of the immune systems of European seabass (*Dicentrarchus labrax*) [27], rainbow trout [28], mandarin fish (*Siniperca scherzeri*) [29], and pearl gentian grouper (*Epinephelus lanceolatus* × *Epinephelus fuscoguttatus*) [30]. Our previous studies showed that bioactive polysaccharides from the melon fly (*Bactrocera cucurbitae*) [5], Japanese oak silkmoth (*Antheraea yamamai*) [6], and silkmoth (*Bombyx mori*) [7] successfully prevent bacterial diseases via the activation of innate immunity. In the present study, we isolated a novel bioactive polysaccharide from the larvae of BSF (*Hermetia illucens*) that can activate innate immune response in mouse RAW264.7 macrophages. Furthermore, FPLC analysis revealed that several separated-fractions from the crude polysaccharides of BSF larvae were found to induce NO production in vitro, indicating the existence of another substance with therapeutic potential for further identification. Hence, the polysaccharide contained in the BSF larvae might play a role in enhancing immunity or altering the gut microbiota in fish or animals.

Innate immunity is widely known as the first line of defense against pathogenic microorganisms and viral infections. Macrophages, together with monocytes and granulocytes, are the main components of the innate immune system and are constantly being used as the main targets to describe immune system activation via the stimulation of immunomodulatory polysaccharides [3,23,31,32]. Among these components, macrophages seem to attract more interest due to their diverse functionality, not only in immunity, but also in other functional activities, such as development, homeostasis, and tissue repair [33]. Macrophages can eliminate foreign matters directly via phagocytosis and indirectly via the induction of numerous macrophage-derived biological factors, such as production of NO and reactive oxygen species, secretion of proinflammatory cytokines, including *TNF-α*, *IL-1β*, *IL-6*, *IL-8*, *IL-12*, and activation of *IFNβ* [1,3,31,34,35]. In our current study, we found a novel immune activator in the extracts of BSF larvae (*Hermetia illucens*) that can stimulate the production of NO and secretion of proinflammatory cytokines in the same manner as that of LPS, a widely known immunomodulator. NO production in mouse macrophage-derived RAW264.7 cells was used as a marker of immunomodulatory activity to asses this newly identified compound. We named this water-soluble immune activator dipterose-BSF.

The activation of innate immunity by macrophages is induced by the recognition of pathogen-associated molecular patterns (PAMPs) in lipids, peptides, nucleic acids, proteins, and carbohydrates in microorganisms [36,37]. Moreover, chitin and chitin derivatives, which are mainly found in the exoskeleton of insects and are comprised of *N*-acetyl-β-d-glucosamine, can also stimulate innate immunity due to their existence in the cell walls of pathogenic fungi [38]. Although a wide variety of bioactive substances and peptides can be found in insect species, Ohta et al. [5] revealed that insects could induce novel immune activators that are not chitin, chitin derivatives, proteins, or peptides. Further analysis using ultrafiltration and proteinase treatment indicated that the novel immune activator from insects was a polysaccharide [6,7]. In our present study, we identified a novel acidic polysaccharide in the BSF larvae *(Hermetia illucens*), termed dipterose-BSF, with a molecular weight of 1.47 × 10^5^ and composed of ten monosaccharides.

Generally, the immunomodulatory activities of polysaccharides are strongly related with their structural features, such as molecular weight, monosaccharide composition, and glycosidic bonds [23,32]. Previous studies suggest that polysaccharides with a high molecular mass usually exhibit significant biological activities compared with low-molecular-mass polysaccharides, as the high-molecular-mass polysaccharides may exhibit more repetitive structures, which could interact with receptors or other membrane targets [39]. The immunomodulatory polysaccharides from insects, such as silkrose of *Antheraea yamamai* (Silkrose-AY) [6], silkrose of *Bombyx mori* (Silkrose-BM) [7], and dipterose of *Bactrocera cucurbitae* (dipterose-BC) [5] were found to be high-molecular-weight polysaccharides, suggesting a positive correlation between high molecular weight and immunomodulatory activity of the bioactive polysaccharides. Similar to previous findings, the current study showed that dipterose-BSF has a high molecular weight (1.47 × 10^5^) and induced the activation of innate immunity in RAW264.7 cells in the same manner as that of LPS.

As described above, polysaccharides with medicinal properties rely on their structure to induce the activation of the immune system. In our current study, we did not show the glycosidic bond of dipterose-BSF. However, monosaccharide analysis by GC-MS system revealed that dipterose-BSF predominantly contains d-galactose, d-glucose, and l-arabinose. Previous studies have proven that the presence of galactose, glucose, and arabinose as monosaccharides can stimulate immunomodulatory activities in RAW264.7 macrophage cell lines [23,32,39], YAC-1 lymphoma cell lines [32], and mice splenocytes [32,39,40]. Besides, dipterose-BSF also possesses l-rhamnose, an activated monosaccharide that mainly exists in plants, fungi, or bacteria [41]. Although insects, in contrast to plants, cannot synthesize uridine diphosphate (UDP)-rhamnose via the de novo pathway of UDP-d-glucose [42], Ohta et al. [5,6] suggested that insects can produce UDP-rhamnose using a de novo pathway from another unidentified UDP-sugar. Therefore, further investigation in this matter is needed to get a better insight into how insects can produce bioactive polysaccharides. Thus, our results suggest that the immunomodulatory activities of dipterose-BSF are closely related to its specific monosaccharides, which may serve as recognition sites of cell receptors, leading to the activation of immune response.

The initial step in the activation of innate immunity is the recognition of unique chemical structures of foreign matters, known as PAMPs, by PRRs [43,44]. One of the PRRs that have often been associated with the stimulation of immune responses by polysaccharides is TLRs [3,43,45]. To date, 12 distinct TLR family members have been found in mammals [46]. Among the TLR members, TLR2 and TLR4 have proven to be important PRRs involved in the recognition of PAMPs induced by bacteria, fungi, parasites, and viruses [37,47]. Naturally derived polysaccharides, such as those from *Ganoderma lucidum* [48], *Pseudallescheria boydii* [49], *Phoma herbarum* [50], and *Lepidium meyenii* [51] have been found to be associated with TLR2 and TLR4 receptors for inducing immunomodulatory activities. In the current study, neutralizing antibodies against TLR2 and TLR4 significantly attenuated the NO-producing activity of dipterose-BSF. Additionally, it is notable that, in this study, dipterose-BSF stimulated both TLR2 and TLR4, in contrast to our previous findings in dipterose-BC that exhibited recognition by TLR4 but not TLR2 [5].

The stimulation of TLR2/4 receptor facilitates the activation of several downstream signaling molecules, such as the adaptor molecule MyD88, IL-1R-associated kinases, transforming growth factor-β-activated kinase (TAK1), TAK1-binding protein 1 (TAB1), TAB2, and TNF receptor-associated factor 6. The subsequent interaction between these signaling molecules plays critical roles in the release of NF-κB into the nucleus, triggering the transcriptional activation of proinflammatory cytokines and production of NO via the secretion of iNOS protein [37,43]. Secretion of cytokines, such as *TNF-α*, *IL-6*, and *IL-1β* is specifically mediated by MyD88-dependent pathways, whereas *IFNβ* is secreted by the activation of *IRF3* via an MyD88-independent pathway [22,52,53]. In our research, treatment of RAW264.7 cells with dipterose-BSF significantly increased the expression levels of not only the proinflammatory cytokines, such as *TNF-α*, *IL-6*, and *IL-1β*, but also those of *IFNβ*, indicating that dipterose-BSF is involved in both MyD88-dependent and -independent pathways to generate proinflammatory cytokines. Moreover, stimulation with dipterose-BSF induced the degradation of IκB and translocation of NF-κB into the nucleus. Taken together, our results demonstrate that TLR2 and TLR4 were the possible receptors of dipterose-BSF in RAW264.7 cells. However, TLR is at least one of many receptors involved in the activation of macrophages, indicating that some other receptors may also be associated with dipterose-BSF stimulation [43].

## 5. Conclusions

In summary, we successfully identified a novel polysaccharide from BSF larvae, named dipterose-BSF, which can activate the innate immunity of mammalian macrophages. Our present study provided us with a wider perspective of the utilization of insects, not only for protein sources, but also as promising functional feed ingredients for further implementation in poultry, livestock, and fish farming. Although the mass-producing techniques of BSF have been well developed, optimization of the purification methods of dipterose-BSF is still needed to achieve wider implementation at an industrial scale. Further investigation of dipterose-BSF by in vivo analysis is also necessary to elucidate its optimal dose for practical application.

## Figures and Tables

**Figure 1 biomolecules-09-00677-f001:**
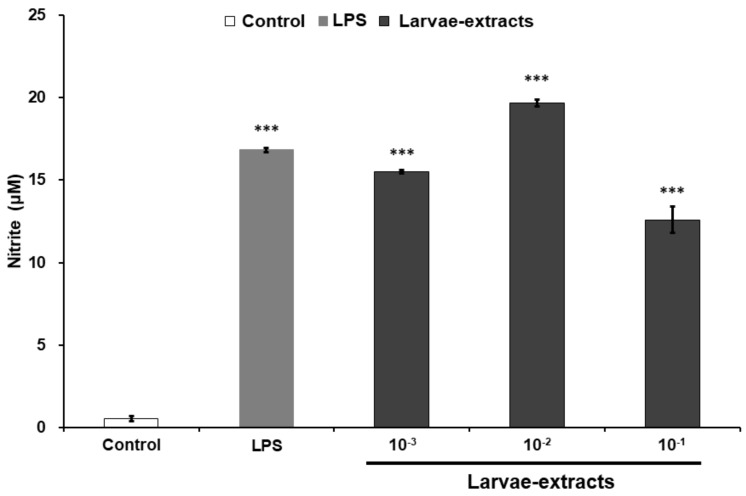
The extracts of black soldier fly (BSF) larvae stimulate nitric oxide (NO) production in RAW264.7 cells. RAW264.7 cells were treated with culture medium alone (control), lipopolysaccharide (LPS; 100 ng/mL), or various concentrations of BSF larvae extract and incubated for 20 h. The levels of nitrite in the culture medium were measured using the Griess assay, as described in the Materials and Methods. All experiments were run in triplicate. Results are expressed as means ± standard error of mean (SEM). Vertical bars indicate SEM of four samples (*n* = 4). *** *p* < 0.001 versus the control group.

**Figure 2 biomolecules-09-00677-f002:**
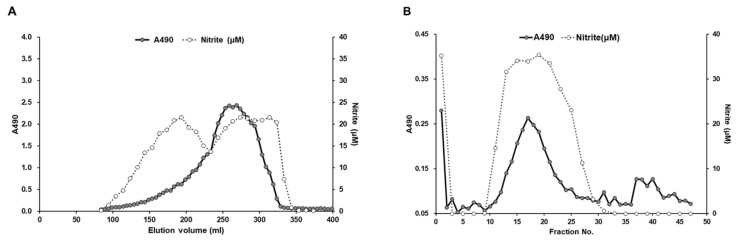
Chromatography profiles of crude polysaccharides in the BSF larvae extract. (**A**) Gel-filtration chromatography profile of the crude polysaccharide fraction of BSF larvae extract. Crude polysaccharides were applied to a Sephacryl S-500HR column. Resulting fractions were collected and the sugar content was measured using the phenol-sulfuric acid method with glucose as a standard (filled circle, optical density (OD) 490 nm). RAW264.7 cells were stimulated with these diluted fractions for 20 h, and the levels of nitrite in the culture medium were measured by the Griess assay (open circle, OD 540 nm). (**B**) Anion-exchange chromatography profile of the fractions with NO-producing ability in RAW264.7 cells on a HiPrep DAEA FF 16/10 column eluted with linear gradient NaCl solution.

**Figure 3 biomolecules-09-00677-f003:**
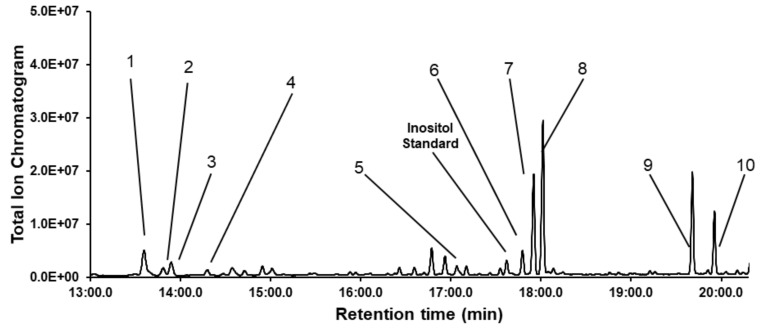
Gas chromatography-mass spectrometry (GC-MS) profile of dipterose-BSF monosaccharides with acid hydrolysis and acetylation. Peaks: 1, l-rhamnose; 2, l-fucose; 3, l-arabinose; 4, d-xylose; 5, d-glucuronic acid; 6, d-mannose; 7, d-glucose; 8, d-galactose; 9, *N*-acetyl-d-glucosamine; 10, *N*-acetyl-d-galactosamine. Inositol represents myo-inositol, an internal standard used for normalization.

**Figure 4 biomolecules-09-00677-f004:**
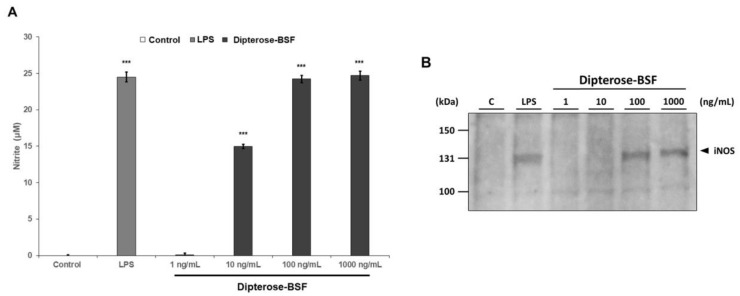
Stimulation with dipterose-BSF induces NO production via the activation of iNOS protein in RAW264.7 cells. (**A**) RAW264.7 cells were stimulated with culture medium alone (control), LPS (100 ng/mL), or various concentrations of dipterose-BSF for 20 h. The nitrite concentrations in the culture medium were then measured using the Griess assay as described in the Materials and Methods. All experiments were run in triplicate (**B**) RAW264.7 cells were incubated with culture medium alone (control), LPS (100 ng/mL), or various concentrations of dipterose-BSF for 6 h. The expression level of iNOS protein was monitored by immunoblot analysis using a rabbit anti-mouse iNOS antibody. Immunoblot image shown here is representative of triplicate. Results are expressed as means ± SEM. Vertical bars indicate SEM of three samples (*n* = 3). *** *p* < 0.001 versus the control group.

**Figure 5 biomolecules-09-00677-f005:**
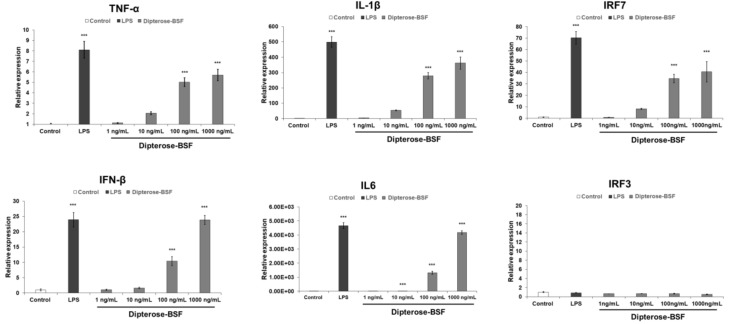
Treatment with dipterose-BSF induces the expression of cytokines and interferon regulatory factor 7 (*IRF7*) in macrophages. RAW264.7 cells were stimulated with culture medium alone (control), LPS (100 ng/mL), or various concentrations of dipterose-BSF. The mRNA expression of the indicated cytokines, *IRF3*, and *IRF7* was measured using real-time PCR analysis at 6 h after treatment with culture medium, dipterose-BSF or LPS. All experiments were run in triplicate, and the results represent the means ± SEM of four samples for each gene analyzed. Vertical bars indicate SEM. *** *p* < 0.001 as compared with the control group.

**Figure 6 biomolecules-09-00677-f006:**
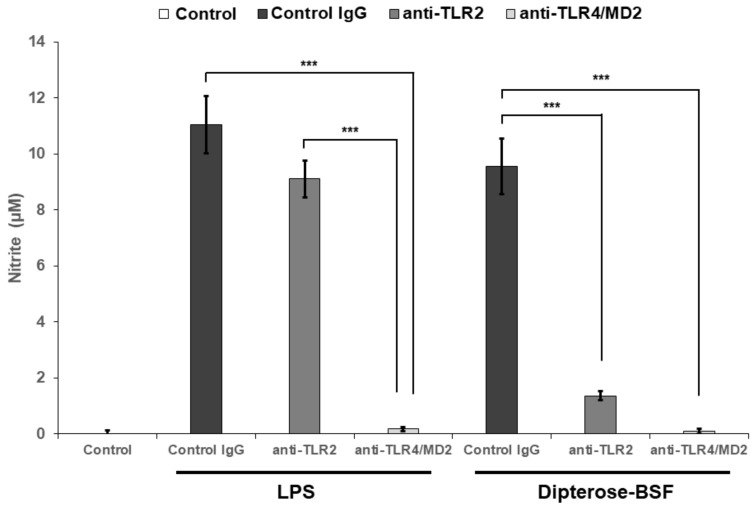
Dipterose-BSF induces NO production in macrophages via the Toll-like receptor (TLR) signaling pathway. Neutralizing antibodies against TLR2 and TLR4 (10 μg/mL) or isotype control IgG (control IgG) were applied to mouse macrophage-derived RAW264.7 cells for 30 min, followed by 20 h of incubation with dipterose-BSF or LPS. Cells in control group were untreated neither with neutralizing antibodies nor stimulants (dipterose-BSF or LPS). The levels of nitrite in the culture medium were then measured using the Griess assay. All experiments were run in triplicate. The results are expressed as means ± SEM. Vertical bars indicate SEM of six samples (*n* = 6). *** *p* < 0.001 versus the control IgG group.

**Figure 7 biomolecules-09-00677-f007:**
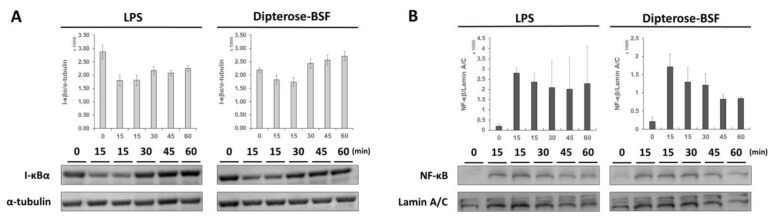
Dipterose-BSF induces the activation of NF-κB via the degradation of IκBα in macrophages. (**A**) RAW264.7 cells were treated with dipterose-BSF (100 ng/mL) or LPS (100 ng/mL) for 0, 15, 30, 45, or 60 min. The expression level of IκBα was detected using immunoblot analysis. α-tubulin served as a protein loading control. The same blot was cut and blotted using anti-mouse α-tubulin antibody. (**B**) RAW264.7 cells were incubated with dipterose-BSF (100 ng/mL) or LPS (100 ng/mL) for 0, 15, 30, 45, or 60 min. The expression level of NF-κB was monitored using immunoblot analysis. Lamin A/C was used as a protein loading control. The same blot was stripped and re-blotted using anti-lamin A/C antibody. Immunoblot images shown here are representative of two independent experiments. Data were expressed as means ± SEM of two samples. Quantitative analysis of protein level on the blot was calculated using ImageJ software (version 1.52a).

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
