# Peer review of "The Dipterose of Black Soldier Fly (Hermetia illucens) Induces Innate Immune Response through Toll-Like Receptor Pathway in Mouse Macrophage RAW264.7 Cells"

_biomolecules, 2019, doi:10.3390/biom9110677_

Round 1

Reviewer 1 Report

The authors identified and isolated a new bioactive polysaccharide from the larvae of BSF. As in the previous studies done by the authors, this bioactive compound induces the innate immune response in mouse macrophage via the TLR pathway. In addition to the potential of using insect as feed ingredients for animal, this species have an added value which could be used a functional feed ingredients. The manuscript is overall well written, with a clear description of the material and methods and particularly interesting to read and provide useful information for the scientific community. This manuscript falls into the scope of the journal, I have only some minor comments:

L22: in vitro 

L58: rainbow trout (Oncorhynchus mykiss)

L59: the study in reference 17 is done on rainbow trout and not Atlantic salmon

L121, 139 and 164: N2, 106 cells and 106), respectively

L188: mean ±                        

Figure 1: is it pupae or larvae?

L 201 with various … or without BSF larvae (control group, cont) also in the figure in addition to the code color for LPS and Pupae-extracts, could you add the definition for Cont

L212: check the reference 5

Figure 4: Values are means (n=?) with their SEM represented by vertical bars.

I am not sure if it is because of the image quality but the expression level of the protein INOS is weak, also did you quantify the protein expression?

All the genes names in italics.

Figure 5, Cont as in Figure 1 or control as in Figure 4 and 5. Could you add the code color for Cont, LPS and dipterose-BSF

Figure 6: could you specify Medium?

Figure 7: could you quantify the bands detected in those western blot?

L329: European seabass (Dicentrarchus labrax)

L337: in vitro

L387: … can stimulate immunomodulatory activities… in which species?

L391: de novo pathway from?

L432; in vivo

Author Response

Specific responses to the reviewers

We would like to express our appreciation to the reviewers for their constructive comments, which have helped us significantly improve the manuscript.

Reviewer comments:

Reviewer #1 (Comments and Suggestions for Authors):

The authors identified and isolated a new bioactive polysaccharide from the larvae of BSF. As in the previous studies done by the authors, this bioactive compound induces the innate immune response in mouse macrophage via the TLR pathway. In addition to the potential of using insect as feed ingredients for animal, this species have an added value which could be used a functional feed ingredients. The manuscript is overall well written, with a clear description of the material and methods and particularly interesting to read and provide useful information for the scientific community. This manuscript falls into the scope of the journal, I have only some minor comments.

Response to Comments and Suggestions for Authors:

We thank the reviewer for these comments. We have addressed the concerns of the reviewer as outlined in our specific responses below.

Detailed comments:

P1L22: in vitro

Response to P1L22: We have made this change.

P2L58: rainbow trout (Oncorhynchus mykiss)

Response to P2L58: We have added “(Oncorhynchus mykiss)”.

P2L59: the study in reference 17 is done on rainbow trout and not Atlantic salmon

Response to P2L59: We have corrected this accordingly as indicated below.

Section

Contents of revision

Introduction paragraph 3

Before

Dietary inclusion of BSF was reported to alter gut microbiota, including increased bacterial diversity to improve gut health in rainbow trout (Oncorhynchus mykiss) [15,16], increased levels of Carnobacterium genus, which is known to act as a probiotic for microbial communities in Atlantic salmon (Salmo salar) [17] and poultry [18].

After

Dietary inclusion of BSF was reported to alter gut microbiota, including increased bacterial diversity to improve gut health in rainbow trout (Oncorhynchus mykiss) [15,16], increased levels of Carnobacterium genus, which is known to act as a probiotic for microbial communities in rainbow trout (Oncorhynchus mykiss) [17] and poultry [18].

P3L121, P4L139, P4L164: N2, 106 cells and 106, respectively

Response to P3L121, P4L139, and P4L164: We have corrected “N2” to “N2”, “106 cells” to “106 cells” and “106” to “106”, respectively.

P5L188: mean ±

Response to P5L188: We have done so.

Figure 1: is it pupae or larvae?

Response to Figure 1: It is larvae, we are sorry for the mistake. We have amended “pupae extracts” to “larvae extracts” as indicated in the next comment’s response below.

P5L201: with various … or without BSF larvae (control group, cont) also in the figure in addition to the code color for LPS and Pupae-extracts, could you add the definition for Cont

Response to P5L201: Accordingly, we have made the revisions and included the definition of control group as indicated below.

Section

Contents of revision

Result

Figure 1

Before

”Please see the attached word file.”

Figure 1. The extracts of BSF larvae stimulate NO production in RAW264.7 cells. RAW264.7 cells were treated with various concentrations of BSF larvae extract and incubated for 20 h. The levels of nitrite in the culture medium were measured using the Griess assay, as described in the Materials and Methods. Results are expressed as means ± standard error of mean (SEM). Vertical bars indicate SEM. ***P < 0.001 versus the control group.

After

”Please see the attached word file.”

Figure 1. The extracts of BSF larvae stimulate NO production in RAW264.7 cells. RAW264.7 cells were treated with culture medium alone (control), LPS (100 ng/mL), or various concentrations of BSF larvae extract and incubated for 20 h. The levels of nitrite in the culture medium were measured using the Griess assay, as described in the Materials and Methods. All experiments were run in triplicate. Results are expressed as means ± standard error of mean (SEM). Vertical bars indicate SEM of four samples (n = 4). ***P < 0.001 versus the control group.

P6L212: check the reference 5

Response to P6L212: We have corrected the reference’s format accordingly.

P7Figure 4: Values are means (n=?) with their SEM represented by vertical bars.

I am not sure if it is because of the image quality but the expression level of the protein INOS is weak, also did you quantify the protein expression?

Response to P7Figure 4: We now included the number of sample (n) in the figure caption. To provide clarity, we have changed the image with same blot’s image but with longer exposure than the previous image. Although we did three independent experiments for the immunoblotting of the protein iNOS, we cannot provide the quantification of protein level due to the absence of positive control for normalization. Our revisions for figure 4 and its figure caption are indicated below.

Section

Contents of revision

Result

Figure 4

Before

”Please see the attached word file.”

Figure 4. Stimulation with dipterose-BSF induces NO production via the activation of iNOS protein in RAW264.7 cells. (A) RAW264.7 cells were stimulated with various concentrations of dipterose-BSF or LPS for 20 h. The nitrite concentrations in the culture medium were then measured using the Griess assay as described in the Materials and Methods. (B) RAW264.7 cells were incubated with various concentrations of dipterose-BSF or LPS for 6 h. The expression level of iNOS protein was monitored by immunoblot analysis using a rabbit anti-mouse iNOS antibody. Results are expressed as means ± SEM. Vertical bars indicate SEM. ***P < 0.001 versus the control group.

After

”Please see the attached word file.”

 Figure 4. Stimulation with dipterose-BSF induces NO production via the activation of iNOS protein in RAW264.7 cells. (A) RAW264.7 cells were stimulated with culture medium alone (control), LPS (100 ng/mL), or various concentrations of dipterose-BSF for 20 h. The nitrite concentrations in the culture medium were then measured using the Griess assay as described in the Materials and Methods. All experiments were run in triplicate. (B) RAW264.7 cells were incubated with culture medium alone (control), LPS (100 ng/mL), or various concentrations of dipterose-BSF for 6 h. The expression level of iNOS protein was monitored by immunoblot analysis using a rabbit anti-mouse iNOS antibody. Immunoblot image shown here is representative of triplicate. Results are expressed as means ± SEM. Vertical bars indicate SEM of three samples (n = 3). ***P < 0.001 versus the control group.

Whole paper sections: All the genes names in italics.

Response: We have corrected this.

P8Figure 5: Cont as in Figure 1 or control as in Figure 4 and 5. Could you add the code color for Cont, LPS and dipterose-BSF

Response to P8Figure 5: Control in figure 5 is RAW264.7 cells treated without LPS or dipterose-BSF (medium alone). We now provide these details in the figure’s caption as well as adding the code color for control, LPS, and dipterose-BSF as indicated below.

Section

Contents of revision

Result

Figure 5

Before

”Please see the attached word file.”

Figure 5. Treatment with dipterose-BSF induces the expression of cytokines and IRF7 in macrophages. RAW264.7 cells were stimulated with various concentrations of dipterose-BSF or LPS. The mRNA expression of the indicated cytokines, IRF3, and IRF7 was measured using real-time PCR analysis at 6 h after treatment with dipterose-BSF or LPS. All samples were run in triplicate, and the results represent the means ± SEM of three samples for each gene analyzed. Vertical bars indicate SEM. ***P < 0.001 as compared with the control group.

After

”Please see the attached word file.”

        Figure 5. Treatment with dipterose-BSF induces the expression of cytokines and IRF7 in macrophages. RAW264.7 cells were stimulated with culture medium alone (control), LPS (100 ng/mL), or various concentrations of dipterose-BSF. The mRNA expression of the indicated cytokines, IRF3, and IRF7 was measured using real-time PCR analysis at 6 h after treatment with culture medium, dipterose-BSF or LPS. All experiments were run in triplicate, and the results represent the means ± SEM of four samples for each gene analyzed. Vertical bars indicate SEM. ***P < 0.001 as compared with the control group.

P9Figure 6: could you specify Medium?

Response to P9Figure 6: Medium is RAW264.7 cells treated without neutralizing antibodies or stimulants (dipterose-BSF or LPS). To provide clarity for the reader, we have changed “medium” to “control” and included description in the figure’s caption as indicated below.

Section

Contents of revision

Result

Figure 6

Before

”Please see the attached word file.”

Figure 6. Dipterose-BSF induces NO production in macrophages via the TLR signaling pathway. Neutralizing antibodies against TLR2 and TLR4 (10 μg/mL) or isotype control IgG were applied to mouse macrophage-derived RAW264.7 cells for 30 min, followed by 20 h of incubation with dipterose-BSF or LPS. The levels of nitrite in the culture medium were then measured using the Griess assay. The results are expressed as means ± SEM. Vertical bars indicate SEM. ***P < 0.001 versus the control IgG group.

After

”Please see the attached word file.”

Figure 6. Dipterose-BSF induces NO production in macrophages via the TLR signaling pathway. Neutralizing antibodies against TLR2 and TLR4 (10 μg/mL) or isotype control IgG (control IgG) were applied to mouse macrophage-derived RAW264.7 cells for 30 min, followed by 20 h of incubation with dipterose-BSF or LPS. Cells in control group were untreated neither with neutralizing antibodies nor stimulants (dipterose-BSF or LPS). The levels of nitrite in the culture medium were then measured using the Griess assay. All experiments were run in triplicate. The results are expressed as means ± SEM. Vertical bars indicate SEM of six samples (n = 6). ***P < 0.001 versus the control IgG group.

P10Figure 7: could you quantify the bands detected in those western blot?

Response to P10Figure 7: We now provide the quantitative analysis of protein level for immunoblot image in figure 7 using ImageJ software. However, we did not conduct statistical analysis for the quantitative analysis result due to the limitation of sample number (n = 2). Experiments for figure 7 were run in two independent experiments. Triplicate independent experiments or more for figure 7 would be difficult due to materials limitation. Our revisions in this regards are indicated in the table below.

Section

Contents of revision

Result

Figure 7

Before

”Please see the attached word file.”

Figure 7. Dipterose-BSF induces the activation of NF-κB via the degradation of IκBα in macrophages. (A) RAW264.7 cells were treated with dipterose-BSF or LPS for 0, 15, 30, 45, or 60 min. The expression level of IκBα was detected using immunoblot analysis. α-tubulin served as a protein loading control. The same blot was cut and blotted using anti-mouse α-tubulin antibody. Full original blot is presented in Supplementary figure S2. (B) RAW264.7 cells were incubated with dipterose-BSF or LPS for 0, 15, 30, 45, or 60 min. The expression level of NF-κB was monitored using immunoblot analysis. Lamin A/C was used as a protein loading control. The same blot was stripped and re-blotted using anti-lamin A/C antibody.

After

”Please see the attached word file.”

Figure 7. Dipterose-BSF induces the activation of NF-κB via the degradation of IκBα in macrophages. (A) RAW264.7 cells were treated with dipterose-BSF (100 ng/mL) or LPS (100 ng/mL) for 0, 15, 30, 45, or 60 min. The expression level of IκBα was detected using immunoblot analysis. α-tubulin served as a protein loading control. The same blot was cut and blotted using anti-mouse α-tubulin antibody. (B) RAW264.7 cells were incubated with dipterose-BSF (100 ng/mL) or LPS (100 ng/mL) for 0, 15, 30, 45, or 60 min. The expression level of NF-κB was monitored using immunoblot analysis. Lamin A/C was used as a protein loading control. The same blot was stripped and re-blotted using anti-lamin A/C antibody. Immunoblot images shown here are representative of two independent experiments. Data were expressed as means ± SEM of two samples. Quantitative analysis of protein level on the blot was calculated using ImageJ software.

P10L329: European seabass (Dicentrarchus labrax)

Response to P10L329: We have added “(Dicentrarchus labrax)”.

P10L337: in vitro

Response to P10L337: We have corrected this.

P11L387: … can stimulate immunomodulatory activities… in which species?

Response to P11L387: We now define the species as indicated below.

Section

Contents of revision

Discussion paragraph 5

Before

Previous studies have proven that the presence of galactose, glucose, and arabinose as monosaccharides can stimulate immunomodulatory activities [23,32,39,40].

After

Previous studies have proven that the presence of galactose, glucose, and arabinose as monosaccharides can stimulate immunomodulatory activities in RAW264.7 macrophage cell lines [23,32,39], YAC-1 lymphoma cell lines [32], and mice splenocytes [32,39,40].

P11L391: de novo pathway from?

Response to P11L391: De novo pathway from unidentified UDP-sugars. Normally, UDP-rhamnose is synthesized through a de novo pathway from UDP-D-glucose in plant. In contrast to plant, insects are unable to synthesize NDP-rhamnose. At this state, we assume that the presence of L-rhamnose in the dipterose-BC (Ohta et. al., 2014) and -BSF was induced by the plant consumed by the insects. However, our previous study in melon fly (Ohta et. al., 2014) suggested that the dipterose-BC levels increased in the pupal stages even though pupae do not feed. This result suggests that the melon fly independently (without eating plant) synthesizes UDP-rhamnose through a de novo pathway of unknown UDP-sugars and then use these products as substrates for the synthesis of dipterose-BC. To date, we are still working to identify these UDP-sugars to get a better understand into how insects can produce bioactive polysaccharide.

To give clarity for the reader, we revised the sentence as indicated below.

Section

Contents of revision

Discussion paragraph 5

Before

Although insects, in contrast to plants, cannot synthesize UDP-rhamnose via the de novo pathway of UDP-D-glucose [42], Ohta et al. [5,6] suggested that insects can produce UDP-rhamnose using another unidentified UDP-sugar through a de novo pathway.

After

Although insects, in contrast to plants, cannot synthesize UDP-rhamnose via the de novo pathway of UDP-D-glucose [42], Ohta et al. [5,6] suggested that insects can produce UDP-rhamnose using a de novo pathway from another unidentified UDP-sugar.

P12L432: in vivo

Response to P12L432: We have corrected this.

Reviewer 2 Report

The authors isolated the activates dipterose-BSF from black soldier fly, which can activate the RAW264.7 through TLR2/4 and NFkB pathway. While, for the most part, the experiments are well conceived and the conclusions appropriate, several concern remain, which need to be addressed.

P4, line 137-143, what dosage of the LPS that used in the RAW 264.7 cell stimulated? P4, line 164, the 1x106 cells are the total cells used in the immunoblot analysis? P5, Figure 1, the results are expressed as means + SEM, however, There are not described in the article. How many independent experiments were test in this study? The same as in the fig4, fig6 and fig7.

Author Response

Specific responses to the reviewers

We would like to express our appreciation to the reviewers for their constructive comments, which have helped us significantly improve the manuscript.

Reviewer comments:

Reviewer #2 (Comments and Suggestions for Authors):

The authors isolated the activates dipterose-BSF from black soldier fly, which can activate the RAW264.7 through TLR2/4 and NFkB pathway. While, for the most part, the experiments are well conceived and the conclusions appropriate, several concern remain, which need to be addressed.

Response to Comments and Suggestions for Authors:

We thank the reviewer for these comments. We have addressed the concerns of the reviewer as outlined in our specific responses below.

Detailed comments:

P4L137-143: what dosage of the LPS that used in the RAW 264.7 cell stimulated?

Response to P4L137-143: We have added the dosage of LPS that used in the RAW264.7 cell stimulated as indicated below.

Section

Contents of revision

Materials and methods

Before

Briefly, cells were seeded at 1 × 106 cells/well in a 96-well plate, pre-incubated for 90 min, and stimulated with varying concentrations of larvae extract or purified polysaccharide in the cultured medium for 24 h at 37 °C. The absorbance of culture medium supernatant was identified at optical density (OD) 540 nm in microplate reader. The nitrite levels in the culture medium were then measured based on the equation from a NaNO2 standard curve.

After

Briefly, cells were seeded at 1 × 106 cells/well in a 96-well plate, pre-incubated for 90 min, and stimulated with LPS (100 ng/mL), varying concentrations of larvae extract or purified polysaccharide in the cultured medium for 24 h at 37 °C. The absorbance of culture medium supernatant was identified at optical density (OD) 540 nm in microplate reader. The nitrite levels in the culture medium were then measured based on the equation from a NaNO2 standard curve.

P4L164: the 1x106 cells are the total cells used in the immunoblot analysis?

Response to P4L164: We mistyped the concentration of cells here. The cell concentration indicated in the sentence is total cell per well. Instead of “1 ´ 106” we changed to “1 ´ 106/well” to provide clarity as indicated below.

Section

Contents of revision

Materials and methods

Before

To prepare cell lysates, cells (1 × 106) were washed three times with ice-cold phosphate buffered saline (PBS) and lysed by sonicating for 30 sec with the addition of 200 μL of lysis buffer (137 mM NaCl, 10 mM phosphate, 2.7 mM KCl, pH 7.4, 1 mM EDTA, and 1% Triton X-100) supplemented with Halt protease inhibitor cocktail (Thermo Fisher Scientific, Rockford, USA).

After

To prepare cell lysates, cells (1 × 106 cells/well) were washed three times with ice-cold phosphate buffered saline (PBS) and lysed by sonicating for 30 sec with the addition of 200 μL of lysis buffer (137 mM NaCl, 10 mM phosphate, 2.7 mM KCl, pH 7.4, 1 mM EDTA, and 1% Triton X-100) supplemented with Halt protease inhibitor cocktail (Thermo Fisher Scientific, Rockford, USA).

P5Figure 1: the results are expressed as means + SEM, however, There are not described in the article. How many independent experiments were test in this study? The same as in the fig4, fig6 and fig7.

Response to P5Figure 1: We now provide the number of sample (n) in the figure’s caption for figure 1, 4, 6 and 7, respectively. All experiments for figure 1, 4, and 6 were run in three independent experiments. Experiments for figure 7 were run in two independent experiments. Triplicate independent experiments or more for figure 7 would be difficult due to materials limitation. Our revisions in this regards are indicated in the table below.

Section

Contents of revision

Result

Caption of Figure 1

Before

Figure 1. The extracts of BSF larvae stimulate NO production in RAW264.7 cells. RAW264.7 cells were treated with various concentrations of BSF larvae extract and incubated for 20 h. The levels of nitrite in the culture medium were measured using the Griess assay, as described in the Materials and Methods. Results are expressed as means ± standard error of mean (SEM). Vertical bars indicate SEM. ***P < 0.001 versus the control group.

After

Figure 1. The extracts of BSF larvae stimulate NO production in RAW264.7 cells. RAW264.7 cells were treated with culture medium alone (control), LPS (100 ng/mL), or various concentrations of BSF larvae extract and incubated for 20 h. The levels of nitrite in the culture medium were measured using the Griess assay, as described in the Materials and Methods. All experiments were run in triplicate. Results are expressed as means ± standard error of mean (SEM). Vertical bars indicate SEM of four samples (n = 4). ***P < 0.001 versus the control group.

Section

Contents of revision

Result

Caption of Figure 4

Before

Figure 4. Stimulation with dipterose-BSF induces NO production via the activation of iNOS protein in RAW264.7 cells. (A) RAW264.7 cells were stimulated with various concentrations of dipterose-BSF or LPS for 20 h. The nitrite concentrations in the culture medium were then measured using the Griess assay as described in the Materials and Methods. (B) RAW264.7 cells were incubated with various concentrations of dipterose-BSF or LPS for 6 h. The expression level of iNOS protein was monitored by immunoblot analysis using a rabbit anti-mouse iNOS antibody. Results are expressed as means ± SEM. Vertical bars indicate SEM. ***P < 0.001 versus the control group.

After

Figure 4. Stimulation with dipterose-BSF induces NO production via the activation of iNOS protein in RAW264.7 cells. (A) RAW264.7 cells were stimulated with culture medium alone (control), LPS (100 ng/mL), or various concentrations of dipterose-BSF for 20 h. The nitrite concentrations in the culture medium were then measured using the Griess assay as described in the Materials and Methods. All experiments were run in triplicate. (B) RAW264.7 cells were incubated with culture medium alone (control), LPS (100 ng/mL), or various concentrations of dipterose-BSF for 6 h. The expression level of iNOS protein was monitored by immunoblot analysis using a rabbit anti-mouse iNOS antibody. Immunoblot image shown here is representative of triplicate. Results are expressed as means ± SEM. Vertical bars indicate SEM of three samples (n = 3). ***P < 0.001 versus the control group.

Section

Contents of revision

Result

Caption of Figure 6

Before

Figure 6. Dipterose-BSF induces NO production in macrophages via the TLR signaling pathway. Neutralizing antibodies against TLR2 and TLR4 (10 μg/mL) or isotype control IgG were applied to mouse macrophage-derived RAW264.7 cells for 30 min, followed by 20 h of incubation with dipterose-BSF or LPS. The levels of nitrite in the culture medium were then measured using the Griess assay. The results are expressed as means ± SEM. Vertical bars indicate SEM. ***P < 0.001 versus the control IgG group.

After

        Figure 6. Dipterose-BSF induces NO production in macrophages via the TLR signaling pathway. Neutralizing antibodies against TLR2 and TLR4 (10 μg/mL) or isotype control IgG (control IgG) were applied to mouse macrophage-derived RAW264.7 cells for 30 min, followed by 20 h of incubation with dipterose-BSF or LPS. Cells in control group were untreated neither with neutralizing antibodies nor stimulants (dipterose-BSF or LPS). The levels of nitrite in the culture medium were then measured using the Griess assay. All experiments were run in triplicate. The results are expressed as means ± SEM. Vertical bars indicate SEM of six samples (n = 6). ***P < 0.001 versus the control IgG group.

Section

Contents of revision

Result

Caption of Figure 7

Before

Figure 7. Dipterose-BSF induces the activation of NF-κB via the degradation of IκBα in macrophages. (A) RAW264.7 cells were treated with dipterose-BSF or LPS for 0, 15, 30, 45, or 60 min. The expression level of IκBα was detected using immunoblot analysis. α-tubulin served as a protein loading control. The same blot was cut and blotted using anti-mouse α-tubulin antibody. Full original blot is presented in Supplementary figure S2. (B) RAW264.7 cells were incubated with dipterose-BSF or LPS for 0, 15, 30, 45, or 60 min. The expression level of NF-κB was monitored using immunoblot analysis. Lamin A/C was used as a protein loading control. The same blot was stripped and re-blotted using anti-lamin A/C antibody.

After

Figure 7. Dipterose-BSF induces the activation of NF-κB via the degradation of IκBα in macrophages. (A) RAW264.7 cells were treated with dipterose-BSF (100 ng/mL) or LPS (100 ng/mL) for 0, 15, 30, 45, or 60 min. The expression level of IκBα was detected using immunoblot analysis. α-tubulin served as a protein loading control. The same blot was cut and blotted using anti-mouse α-tubulin antibody. (B) RAW264.7 cells were incubated with dipterose-BSF (100 ng/mL) or LPS (100 ng/mL) for 0, 15, 30, 45, or 60 min. The expression level of NF-κB was monitored using immunoblot analysis. Lamin A/C was used as a protein loading control. The same blot was stripped and re-blotted using anti-lamin A/C antibody. Immunoblot images shown here are representative of two independent experiments. Data were expressed as means ± SEM of two samples. Quantitative analysis of protein level on the blot was calculated using ImageJ software.
